# LLawCo: Learning Laws of Cooperation for Modeling Embodied Multi-Agent Behavior

**Qinhong Zhou** [1] [*]   **Chuang Gan** [1]   **Anoop Cherian** [2]

## Abstract

Embodied agents operating in decentralized and partially observable environments have attracted growing attention in recent years. However, existing large language model (LLM)–based agents often exhibit behaviors that are misaligned with their partners or inconsistent with the environment state, leading to inefficient cooperation and poor task success. To address this challenge, we propose a novel framework, Learning Laws of Cooperation (LLawCo), that enables embodied agents to autonomously align with both their partners and task objectives. Our framework allows agents to reflect on past failures to extract misaligned behavioral patterns, which are used to derive high-level behavioral laws (e.g., *"Talk when necessary"*, *"Wait for partner"*). These laws are explicitly incorporated into the agents' chains of thought via supervised fine-tuning, aligning their reasoning with task requirements and the behavior of other agents. To evaluate our approach, we introduce PARTNR-Dialog, a large-scale multi-agent communicative and cooperative planning benchmark built on the PARTNR environment. Experiments on existing tasks and our new benchmark demonstrate significant improvements in cooperative efficiency and task success rates. Across four backbone LLMs, our method achieves average success rate improvements of 4.5% on the PARTNR-Dialog benchmark and 6.8% on the TDW-MAT benchmark over state-of-the-art open-source communicative agent frameworks.

[*]Work done during an internship at MERL. [1]University of Massachusetts Amherst, USA [2]Mitsubishi Electric Research Laboratories (MERL), USA. Correspondence to: Anoop Cherian <cherian@merl.com>.

*Proceedings of the $43^{rd}$ International Conference on Machine Learning*, Seoul, South Korea. PMLR 306, 2026. Copyright 2026 by the author(s).

## 1. Introduction

Recent years have witnessed growing interest in building frameworks to facilitate cooperation among embodied agents operating in complex, partially observable environments (Puig et al., 2020; Thomason et al., 2020; Puig et al., 2023; Chang et al., 2024). Among various problem settings, *communicative embodied agents* represent one of the most challenging forms of cooperation, as agents must jointly reason, act, and communicate under partial observability. The dominant line of work (Mandi et al., 2024; Zhang et al., 2024b; Liu et al., 2024; Guo et al., 2024; Zu et al., 2025; Ling et al., 2025) relies on large language model (LLM)-based agents, which leverage LLMs for both high-level decision-making and natural language communication. Despite encouraging progress, existing methods suffer from two fundamental limitations.

First, current approaches (Zhang et al., 2024b; Liu et al., 2024; Zu et al., 2025) lack effective mechanisms to align LLM-based agents with both their cooperation partners and task specifications. In realistic cooperative scenarios, an agent is expected to collaborate with diverse partners including humans and heterogeneous robots across varying environments and tasks. Achieving efficient cooperation in such settings requires agents to rapidly adapt to partner-specific behaviors, preferences, and task constraints. However, existing communicative agents often fail to explicitly model and internalize such alignment, leading to suboptimal coordination and degraded task performance.

Second, existing frameworks lack learning-based mechanisms that allow LLM-based agents to *self-evolve*. Most prior works either improve performance through carefully designed non-learning pipelines, which can alleviate certain failure modes but remain fundamentally constrained by the capabilities of the underlying base LLM (Liu et al., 2024; Zu et al., 2025; Ling et al., 2025; Li et al., 2025), or rely on supervision distilled from stronger models to transfer capabilities to weaker ones (Zhang et al., 2024b; Chang et al., 2024). The latter approach requires access to more powerful models and does not enable agents to improve autonomously through interaction. As a result, current agents are unable to continuously adapt and improve in practical, long-horizon cooperative settings.

To address these challenges, we draw inspiration from Isaac Asimov's Three Laws of Robotics (Asimov, 2004), which prescribe high-level laws that agents must adhere to in their behavior. Rather than manually specifying such laws—which can be cumbersome and brittle—we propose Learning Laws of Cooperation (LLawCo), a framework that enables emboided agents to autonomously learn effective behavioral laws. In our learning-based approach, agents reflect on and summarize recurring patterns of past failures, as well as on the constraints enforced by the environment, tasks, and partners into a set of abstract *laws*. Guided by these laws and task performance signals, agents reorganize their chains of thought and selectively filter interaction traces to generate training data. We then embed these laws into the LLM via supervised fine-tuning, enabling the agent to explicitly internalize and adhere to the discovered principles in its task execution and cooperation.

Our LLawCo framework offers several desirable properties. First, the learned laws provide an interpretable and controllable mechanism for alignment, allowing agents to strictly follow discovered regularities. Moreover, by manually specifying or modifying laws, agents can be guided to comply with human preferences or domain-specific priors during cooperation. Crucially, our method does not rely on supervision from stronger models or ground-truth annotations. Instead, agents are able to self-align at the level of their underlying (LLM) reasoning models through interaction, leading to improved cooperation efficiency and task performance.

A key practical impediment to progress for studying communicative cooperation among embodied agents is the absence of suitable benchmarks at scale. Widely used benchmarks such as COELA (Zhang et al., 2024b; Gan et al., 2022) are relatively small, while C-WAH (Puig et al., 2020; 2018) does not provide training data, significantly limiting the development of data-driven, learning-based approaches for communicative embodied agents. To support research in this direction, we introduce **PARTNR-Dialog**, a new large-scale communicative cooperation benchmark built upon the PARTNR environment (Chang et al., 2024; Puig et al., 2023). PARTNR-Dialog extends the original benchmark with a richer set of cooperative actions and communicative interactions. To accommodate the increased scale of the dataset, we further optimize the benchmark framework to significantly improve the efficiency.

We evaluate our LLawCo framework across the PARTNR-Dialog benchmark and the TDW-MAT benchmark using a diverse set of LLM backbones. Experimental results demonstrate that our method consistently and significantly improves both cooperation efficiency and task success rates. Across four different model architectures, our approach achieves an average improvement of 4.5% in success rate over the strongest communicative baseline on PARTNR-Dialog, and an average improvement of 6.8% over the strongest baseline in success rate on TDW-MAT. Furthermore, we show that modifying the laws at inference time leads agents to reliably follow the updated constraints, highlighting the controllability and flexibility of our framework.

Our contributions are summarized as follows:

• We propose a **law-guided learning framework** that enables interpretable, controllable, and continual improvement of communicative embodied agents, without requiring supervision from stronger models or ground-truth annotations.

• We demonstrate consistent performance gains across multiple benchmarks and LLM backbones, with a 14B model achieving performance comparable to a 70B model in certain settings.

• We introduce **PARTNR-Dialog**, a new large-scale benchmark that addresses the data scarcity and scalability limitations of existing communicative embodied agent environments.

## 2. Related Works

**Embodied Communicative Cooperation.** Large-scale embodied AI simulators provide interactive, physically grounded environments for studying intelligent agents (Beattie et al., 2016; Wu et al., 2018; Xiang et al., 2020; Li et al., 2024a; Puig et al., 2018; Kolve et al., 2017; Yan et al., 2018; Makoviychuk et al., 2021). Building upon these platforms, embodied multi-agent cooperation has recently attracted increased attention as an important research problem, focusing on how two or more agents collaborate to complete shared tasks in such environments (Puig et al., 2023; Chang et al., 2024; Puig et al., 2020; Jain et al., 2020; Szot et al., 2023; Zhang et al., 2024a;d;c; Chen et al., 2024a; Zhou et al., 2025). Within this setting, a line of work improves cooperative efficiency by enabling communication among agents. A common approach in this direction is to build agent frameworks using off-the-shelf language models (Mandi et al., 2024; Nayak et al., 2024; Liu et al., 2024; Zu et al., 2025; Ling et al., 2025; Li et al., 2025; Kim et al., 2025), which are responsible for both communication and task execution. To further improve performance, some prior works rely on supervision from stronger models (Zhang et al., 2024b; Chang et al., 2024), collecting high-quality interaction traces and distilling them into smaller models, thereby enhancing communication effectiveness and task success rates.

**Law-Guided Reasoning for LLM-based Agents.** Prior works have explored aligning large language models (LLMs) using high-level principles or rules, rather than relying on dense human supervision. Constitutional AI (Bai et al., 2022; Kundu et al., 2023) and Self-Align (Sun et al.,

2023) demonstrate that a small set of laws can guide self-alignment, enabling LLMs to perform inference with explicitly specified principles. Building on this idea, recent works further investigate making rules more programmable and controllable (Mu et al., 2024; Findeis et al., 2024), self-refining (Wang et al., 2023; Madaan et al., 2023; Chen et al., 2024b; Yuan et al., 2024), and its applications (Li et al., 2024b). In contrast to these approaches, our work focuses on *embodied agents* and leverages laws to model both task structure and partner behaviors in embodied and cooperative settings. By grounding laws in embodied interaction and multi-agent conversations, our method enables more effective coordination and improved task efficiency, going beyond purely text-based alignment objectives.

## 3. Proposed Method

Training effective multi-agent embodied AI systems presents significant data collection challenges, particularly for communicative agents that require high-quality dialog–action pairs. In particular, there are two key challenges: (1) collecting meaningful communication data, which often requires strong model capabilities to generate coherent dialog and well-coordinated actions, and (2) simultaneously achieving task effectiveness while satisfying user preferences when such preferences are present.

We propose Learning Laws of Cooperation (LLawCo) – a law-guided framework that addresses these challenges through failure-driven law extraction and success-driven action alignment, as illustrated in Figure 1. Our method consists of three key components: (1) failure-driven law extraction, (2) success analysis with law-guided explanation, and (3) law-guided agent alignment. The framework enables both *controllability* through human-editable laws and *self-evolve*, allowing LLM-based embodied agents to improve autonomously without requiring supervision from human annotations or stronger models.

### 3.1. Problem Formulation

Consider a multi-agent communicative embodied AI task in which two agents, $A_0$ and $A_1$, collaborate to complete the same task. Each agent is equipped with an LLM backbone $f_\theta$ parameterized by learnable weights $\theta$. An episode $\mathcal{E}$ consists of a sequence of actions $\mathbf{a} = (a_0^t, a_1^t)_{t=1}^T$, observations $\mathbf{o} = (o_0^t, o_1^t)_{t=1}^T$, and communication messages $\mathbf{m} = (m_0^t, m_1^t)_{t=1}^T$ generated by both agents over a horizon of $T$ steps. The task completion rate $\rho(\mathcal{E})$ measures the proportion of task requirements that are successfully satisfied within the episode $\mathcal{E}$. An episode is considered successful if $\rho(\mathcal{E}) \geq \tau$, where $\tau$ is a predefined threshold, and is considered failed otherwise.

### 3.2. Communication-Aware Planning

Our agents are implemented on top of the ReAct agent framework (Yao et al., 2022; Chang et al., 2024). In this framework, a ReAct agent triggers replanning either at the initial step or when feedback from the previous action is received, and then decides the next action to execute. In the LLM-based ReAct agent, the agent relies on a large language model to select the next action at each replanning step. All environment observations, task specifications, and action histories are parsed into natural language and provided as input to the LLM in the form of a structured prompt. In this work, we focus exclusively on LLM-based agents.

To equip LLM-based agents with communication capabilities during planning, we maintain an event history buffer for each agent, which records all messages received from other agents along with their corresponding time steps and source agent identifiers. Following prior works (Zhang et al., 2024b; Kim et al., 2025), the communication infrastructure incorporates the accumulated dialog history into the planner prompt during replanning using a structured template. This design enables the agent to explicitly reason about coordination, task allocation, and partner intentions based on the communication history observed within the episode.

### 3.3. Failure-Driven Law Extraction

Based on the communication-aware planning framework, our goal is to enable agents to autonomously learn a set of laws that align their behavior with both the task objectives and their cooperation partners. To this end, we first learn a law set $\mathcal{P} = \{p_i\}_{i=1}^K$ that guides agent behavior, where each law $p_i = (n_i, d_i)$ consists of a name $n_i$ and a natural language description $d_i$ specifying a behavioral principle. These laws are expressed at a high level and are intended to capture recurring coordination patterns, failure modes, and task-specific constraints observed in interaction traces.

Given a set of failed episodes $\mathcal{E}_{fail} = \{\mathcal{E}_j : \rho(\mathcal{E}_j) < \tau\}$, we extract laws through a two-step process:

**Failure Analysis.** For each failed episode $\mathcal{E}_j \in \mathcal{E}_{fail}$, we extract its full chain of thought reasoning trace $\tau_j$ and use an LLM $f_\theta$ to identify the failure reason:

$$r_j = f_\theta(\tau_j, \mathcal{T}_{failure}), \tag{1}$$

where $\mathcal{T}_{failure}$ is a prompt template that guides the model to analyze communication issues, planning errors, action execution problems, coordination failures, and task understanding mistakes. See Appendix H for prompt details.

The following illustrates a representative failure reason identified by the model:

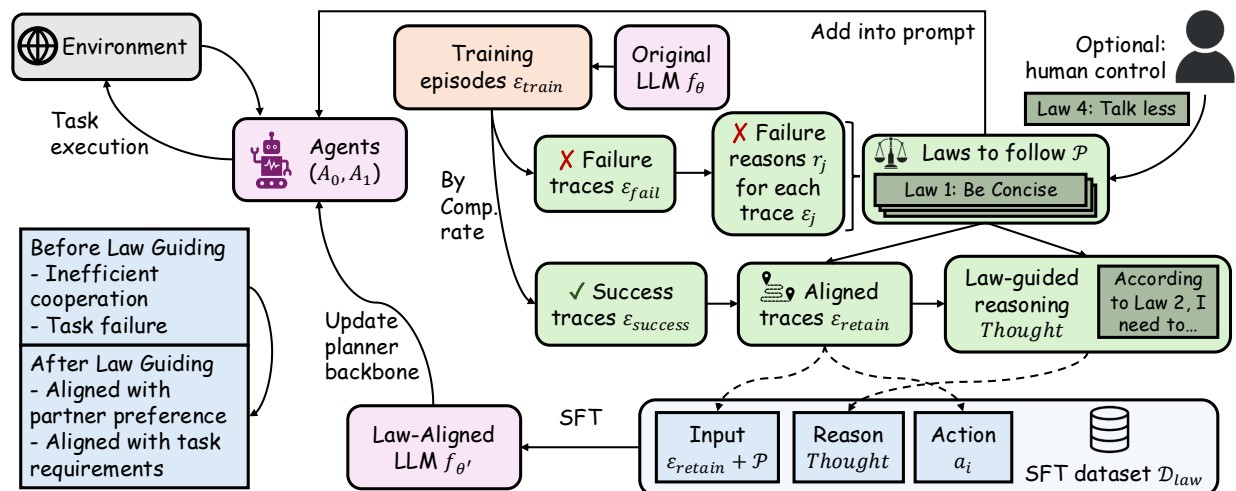

*Figure 1.* **Overview of the proposed LLawCo framework.** Agents interact with the environment to collect training episodes, which are separated into failure traces and success traces based on task completion. Failure traces are analyzed to extract high-level laws, while success traces are filtered to retain law-aligned behaviors. These law-aligned traces are then used to generate law-guided reasoning and actions, forming supervision for fine-tuning the LLM-based planner. During inference, the fine-tuned agent follows the learned or user-specified laws to guide communication, planning, and action selection, enabling explicit human control by editing the laws.

> Failure Reason: Agent 1 repeatedly requested Agent 0 to place objects on the table but did not take action to pick them up themselves, leading to a coordination breakdown.

**Law Summarization.** We aggregate all failure reasons $\mathbf{R} = \{r_j\}_{j=1}^{|\mathcal{E}_{fail}|}$ and generate laws:

$$\mathcal{P} = f_\theta(\mathbf{R}, \mathcal{T}_{law}), \qquad (2)$$

where $\mathcal{T}_{law}$ is a prompt template that instructs the model to generate at most $K$ laws that are clear, actionable, and generalizable, and that address common failure patterns. The structured output $\mathcal{P} = \{(n_i, d_i)\}_{i=1}^{K}$ consists of a set of law tuples, where each tuple pairs a law name $n_i$ with its corresponding description $d_i$. The number of laws $K$ is a hyperparameter. In our experiments, we set $K = 5$, and we provide an ablation study on different choices of $K$ in the experimental section. An example law and its description produced by the above pipeline is provided below. Additional examples of learned laws in $\mathcal{P}$ are provided in Appendix A.

> **Track State Accurately**: Agents should maintain an accurate and consistent representation of the task state, including object locations and statuses.

**Law-Aligned Trace Selection.**

For each successful episode $\mathcal{E}_j \in \mathcal{E}_{success} := \mathcal{E}_{train} \backslash \mathcal{E}_{fail}$ in training episodes $\mathcal{E}_{train}$, we extract the full action histories $\mathbf{a}_j^0$ and $\mathbf{a}_j^1$ of both agents. Our goal is to select episodes

that are both successful and aligned with the learned laws $\mathcal{P}$.

Specifically, we condition the backbone LLM on the law set and the episode trace, and prompt the LLM to determine whether the episode is law-aligned:

$$\epsilon_{align}^{(j)} = f_\theta(\mathbf{a}_j^0, \mathbf{a}_j^1, \mathcal{P}, \mathcal{T}_{align}), \quad \epsilon_{align}^{(j)} \in \{0, 1\}, \quad (3)$$

where $\mathcal{T}_{align}$ is a prompt template that instructs the model to assess the consistency between agent behaviors and the specified laws. Only episodes that are both successful and law-aligned are retained:

$$\mathcal{E}_{retain} = \{\mathcal{E}_j \mid \epsilon_{succ}^{(j)} = 1 \land \epsilon_{align}^{(j)} = 1\}, \qquad (4)$$

where $\epsilon_{succ}^{(j)} = 1$ denotes a successful episode i.e., $\rho(\mathcal{E}_j) \geq \tau$.

### 3.4. Law-Guided Agent Alignment

We align agent behavior through supervised fine-tuning (SFT) on the collected training data, followed by law-guided inference.

**Law-Guided Data Generation.** For the retained episodes $\mathcal{E}_{retain}$, we generate law-guided supervision data from successful executions. For each retained episode $\mathcal{E}_j \in \mathcal{E}_{retain}$, we consider the sequence of successfully executed actions $\mathbf{a}_j = (a_0^t, a_1^t)_{t=1}^{T}$.

For each action $a_k^t \in \mathbf{a}_j$ at time step $t$, we collect the corresponding observation history $\mathbf{o}_k^{\leq t}$ together with the executed action $a_k^t$. We then prompt the backbone LLM to generate a law-guided reasoning trace:

$$\text{Thought}_k^t = f_\theta(\mathbf{o}_k^{\leq t}, a_k^t, \mathcal{P}, \mathcal{T}_{reflect}), \qquad (5)$$

where $\mathcal{T}_{\text{reflect}}$ is a prompt template that requires the model to justify the action choice by explicitly grounding its reasoning in the most relevant law $(n_i, d_i) \in \mathcal{P}$. This single-law grounding design encourages focused reasoning and avoids requiring an explicit multi-law composition module during training. This process produces a law-guided training dataset

$$\mathcal{D}_{\text{law}} = \{(\mathbf{o}_k^{\leq t}, \text{Thought}_k^t, a_k^t)\}, \tag{6}$$

which explicitly links successful actions to their law-based justifications.

**Supervised Fine-Tuning.** We use the collected training dataset $\mathcal{D}_{\text{law}}$ to fine-tune the backbone LLM $f_\theta$:

$$f_{\theta'} = \text{SFT}(f_\theta, \mathcal{D}_{\text{law}}), \tag{7}$$

where $f_{\theta'}$ denotes the fine-tuned model that learns to generate *law-referencing thoughts* together with the corresponding actions. We train the model using a standard cross-entropy loss with next-token prediction supervision. Specifically, the benchmark description is provided in the system prompt, while the observation and task description are included in the user prompt. Supervision is applied on the model outputs, which consist of both the law-guided reasoning trace and the final action.

**Law-Guided Inference.** During inference, we use the fine-tuned model $f_{\theta'}$ and explicitly provide the extracted laws $\mathcal{P}$ in the prompt. At each time step $t$, the agent performs a single autoregressive generation conditioned on the current context $\mathbf{h}^t$ and the law set $\mathcal{P}$. The prompt is constructed using a template $\mathcal{T}_{\text{inference}}$ that incorporates the laws and instructs the model to reason in accordance with them.

Concretely, given the input context $(\mathbf{h}^t, \mathcal{P}, \mathcal{T}_{\text{inference}})$, the model sequentially generates an intermediate law-referencing thought followed by the action:

$$(\text{Thought}_i^t, a_i^t) \sim f_{\theta'}(\mathbf{h}^t, \mathcal{P}, \mathcal{T}_{\text{inference}}), \tag{8}$$

where $\mathbf{h}^t$ summarizes the agent's observation history, action history, and message history up to time step $t$. The thought explicitly references relevant laws from $\mathcal{P}$ and serves as an interpretable justification for the subsequent action. The inference prompt provides the full law set, while the selection of the most relevant law and any tradeoff among applicable laws are learned implicitly from the law-grounded SFT data. This inference procedure ensures that the selected action is directly conditioned on the laws through the generated reasoning, without requiring a separate planning or verification stage.

### 3.5. Implementation

Our framework is implemented as a modular pipeline with six stages: (1) law extraction from $\mathcal{E}_{\text{fail}}$, (2) law-aligned trace $\mathcal{E}_{\text{retain}}$ selection, (3) law-guided action reflection, (5)

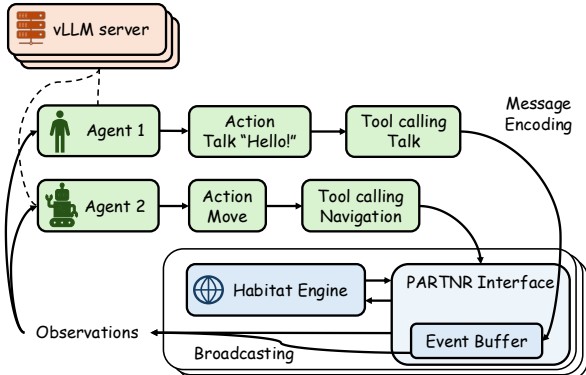

*Figure 2.* **Illustration of the PARTNR-Dialog architecture.** The `Talk` action is implemented via tool calling and handled through per-agent event buffers maintained at the PARTNR interface layer. The buffers route messages to individual agents, after that each agent plans independently from its own observation, history, and received messages. The benchmark is decomposed into three components: an LLM server, a planner, and a simulator. By decoupling LLM inference from simulation and executing the LLM server and simulator in parallel, the framework achieves efficient large-scale evaluation.

supervised fine-tuning, and (6) law-guided inference. As Figure 1 shows, with the laws $\mathcal{P}$ serving as the central interface, LLawCo enables human oversight through direct $\mathcal{P}$ modification.

## 4. Experiments

### 4.1. PARTNR-Dialog Benchmark

Among existing benchmarks for communicative embodied agents, two of the most widely used are C-WAH and TDW-MAT. However, C-WAH does not provide a training split, while TDW-MAT contains only 24 training episodes, significantly limiting learning-based research for communicative embodied agents.

To address this issue, we choose to build upon the PARTNR benchmark (Chang et al., 2024), which provides a substantially larger dataset with 2,000 training episodes and 1,000 validation episodes. Based upon it, we introduce the PARTNR-Dialog Benchmark, an extension that enables explicit communication between agents in multi-agent embodied tasks.

**Communication Infrastructure** As shown in Figure 2, the PARTNR-Dialog Benchmark extends the original PARTNR environment with a communication infrastructure that supports message passing between agents. At the core of this infrastructure is the `Talk` action, a new primitive action that agents can invoke to broadcast messages to other agents in the environment. When an agent executes `Talk[message]`, the message content is encoded and propagated through the event buffer. The environment im-

plements a message encoding scheme using a special separator token to encode both the source agent identifier and the message content. The environment interface maintains an event buffer that collects all speech events generated during each simulation step. These events are encoded as integer tensors, following the same format as other actions to ensure interface consistency. In the subsequent simulation steps, the encoded events are decoded and broadcast to all agents except the speaker. The received messages are included as part of each agent's observation.

**Distributed Inference Acceleration** To enable efficient large-scale evaluation, we implement a client-server architecture that decouples LLM inference from simulator execution. In this framework, the LLM inference is deployed on a GPU server and accelerated using the vLLM framework (Kwon et al., 2023), significantly reducing inference latency. On the client side, where simulators are executed, we exploit the fact that Habitat 3.0 can run multiple simulators in parallel on a single GPU (Puig et al., 2023). We therefore adopt episode-level parallelism to maximize GPU utilization and alleviate the GPU bottleneck during evaluation. For example, when evaluating LLaMA-3.1-8B on the validation set, our framework requires only 4 GPU hours on A100 GPUs under sufficient CPU resources.

### 4.2. Experimental Settings

**Tasks.** We evaluate our method on two communicative embodied multi-agent benchmarks: PARTNR-Dialog and TDW-MAT. **PARTNR-Dialog** is our extension of the PARTNR benchmark that adds explicit communication capabilities, as detailed in Section 4.1. We also conduct experiments on **TDW-MAT** (Zhang et al., 2024b), a multi-agent task benchmark built on the ThreeDWorld (Gan et al., 2020) simulator, which focuses on collaborative manipulation scenarios involving food and household object categories. We do not report results on C-WAH or other communicative benchmarks, as our approach requires fine-tuning the agent models, while these benchmarks do not provide a training split, making them unsuitable for learning-based evaluation.

**Metrics.** For PARTNR-Dialog, following PARTNR (Chang et al., 2024), we report four primary metrics: percentage complete, success rate, steps, and replan rounds. *Percentage complete* (Comp.) measures the proportion of task constraints that are satisfied at the end of the episode. The *success rate* (Succ.) reflects whether all task constraints are satisfied in an episode. *Steps* denote the total number of simulator steps required to complete an episode, and *Replan* indicate the number of times the LLM-based planner triggers replanning. In addition, to provide a more comprehensive evaluation of communication behaviors, we also report the average number of dialog turns per episode, denoted as *Talk*. For TDW-MAT, following CoELA (Zhang

et al., 2024b), we report the success rates on the *food* and *stuff* subtasks, as well as the overall success rate.

**LLM Backbones.** Across all experiments, we evaluate four language model backbones: LLaMA-3.1-8B (Dubey et al., 2024), LLaMA-3.1-70B (Dubey et al., 2024), Gemma-3-12B (Team et al., 2025), and Qwen-3-14B (Yang et al., 2025). To set up the LLM during inference, including selecting the hyperparameters such as maximum tokens, temperature, top-$k$, and top-$p$, we use the corresponding values used in TDW-MAT and COELA (Zhang et al., 2024b) for their TDW-MAT experiments. We use the same settings as in PARTNR (Chang et al., 2024) for our PARTNR-Dialog experiments.

### 4.3. Baselines

**ReAct** (Yao et al., 2022) is a non-communicative planner baseline adopted in PARTNR (Chang et al., 2024). In ReAct, replanning is triggered either at step 0 or after the completion of the previous action. During replanning, the LLM-based planner selects the next action based on an input prompt that includes the task description, environment information, agent states, and action history. Available actions include exploration, navigation, object pickup, and related embodied operations.

**RoCo** (Mandi et al., 2024) is a simple and direct communicative agent baseline. RoCo allows agents to engage in multiple rounds of dialog at the beginning of the episode to discuss high-level strategies. After task execution begins, agents focus solely on acting and no longer communicate. In our experiments, we enforce three rounds of dialog at the start of each episode and then disable the communication action for the remainder of the task.

**CoELA** (Zhang et al., 2024b) is a widely used communicative agent framework. Unlike RoCo, CoELA allows agents to initiate communication at any planning step. Each planning step in CoELA consists of two substeps. In the first substep, an LLM-based communication module generates a message based on the current task state and dialog history. In the second substep, the planning module selects the next action from the available action list, which includes sending the previously generated message, thereby jointly determining both communication and physical actions.

**CommPARTNR** refers to the agent baseline proposed in Communicative-PARTNR (Kim et al., 2025). This baseline extends ReAct by incorporating explicit support for communication. Specifically, during each planning step, the LLM is allowed to select the `talk` action and generate the corresponding message content within the same inference.

*Table 1.* Results on **PARTNR-Dialog**. Each language model is evaluated under four existing planning frameworks (ReAct, RoCo, CoELA, CommPARTNR) and our proposed method (**Ours**).

| Method | Comp. | Succ. | Steps | Replan | Talk |
|---|---|---|---|---|---|
| **LLaMA-3.1-8B** | | | | | |
| ReAct | **0.77** | **0.62** | 3606.2 | **20.7** | 0.0 |
| RoCo | 0.69 | 0.52 | 3940.8 | 29.1 | 3.0 |
| CoELA | 0.72 | 0.51 | 3694.3 | 24.4 | 4.7 |
| CommPARTNR | 0.71 | 0.49 | 3671.3 | 28.6 | 3.6 |
| LLawCo | 0.74 | 0.57 | **3456.0** | 23.0 | 1.9 |
| **LLaMA-3.1-70B** | | | | | |
| ReAct | 0.86 | 0.73 | 3295.2 | **15.2** | 0.0 |
| RoCo | 0.81 | 0.66 | **2407.8** | 19.8 | 3.0 |
| CoELA | 0.80 | 0.65 | 2572.0 | 17.6 | 4.2 |
| CommPARTNR | 0.85 | 0.70 | 3157.5 | 16.9 | 1.6 |
| LLawCo | **0.87** | **0.75** | 2526.5 | 16.1 | 1.9 |
| **Gemma-3-12B** | | | | | |
| ReAct | 0.75 | 0.59 | 3547.1 | 17.2 | 0.0 |
| RoCo | 0.73 | 0.56 | **1837.9** | 21.2 | 3.0 |
| CoELA | 0.68 | 0.42 | 3516.2 | 23.9 | 4.9 |
| CommPARTNR | 0.76 | 0.60 | 3282.9 | 20.5 | 0.9 |
| LLawCo | **0.79** | **0.61** | 2741.6 | **16.0** | 0.9 |
| **Qwen-3-14B** | | | | | |
| ReAct | 0.80 | 0.64 | 2563.1 | 15.4 | 0.0 |
| RoCo | 0.81 | 0.69 | 2491.8 | 25.0 | 3.0 |
| CoELA | 0.71 | 0.62 | 3136.9 | 36.0 | 7.1 |
| CommPARTNR | 0.81 | 0.70 | 3518.5 | 21.0 | 1.2 |
| LLawCo | **0.84** | **0.74** | **2131.8** | **15.1** | 1.6 |

*Table 2.* Results on **TDW-MAT**. The heuristic baseline (MHP) is model-independent. Each language model is evaluated with RoCo, CoELA, and our proposed method (**Ours**).

| Method | Food | Stuff | Total |
|---|---|---|---|
| MHP | 0.76 | 0.74 | 0.75 |
| **LLaMA-3.1-8B** | | | |
| RoCo | 0.42 | 0.32 | 0.37 |
| CoELA | 0.61 | 0.61 | 0.61 |
| LLawCo | **0.71** | **0.63** | **0.67** |
| **LLaMA-3-70B** | | | |
| RoCo | 0.79 | 0.73 | 0.76 |
| CoELA | 0.74 | 0.71 | 0.73 |
| LLawCo | **0.86** | **0.78** | **0.82** |
| **Gemma-3-12B** | | | |
| RoCo | 0.66 | 0.67 | 0.66 |
| CoELA | **0.73** | 0.58 | 0.65 |
| LLawCo | **0.73** | **0.68** | **0.70** |
| **Qwen-3-14B** | | | |
| RoCo | 0.73 | 0.58 | 0.65 |
| CoELA | 0.73 | 0.73 | 0.73 |
| LLawCo | **0.80** | **0.80** | **0.80** |

## 4.4. Results on PARTNR-Dialog

Table 1 reports comprehensive results across all model sizes and communicative baselines. Our method consistently outperforms all communicative baselines across different backbone LLMs, demonstrating the effectiveness and robustness of the proposed law-guided framework. Importantly, our approach achieves performance gains purely through self-alignment of the backbone LLM, instead of relying on supervision from stronger models or external data. This result indicates that our method generalizes well across models of different sizes and architectures, enabling agents to autonomously improve task performance. In particular, our method significantly outperforms all communicative baselines, including RoCo, CoELA, and CommPARTNR.

Across model scales, our method achieves a success rate of 0.74 with Qwen-3-14B, surpassing all baseline methods that employ LLaMA-3.1-70B. This highlights the efficiency of our framework in enabling smaller or mid-sized models to match or exceed the performance of substantially larger models. For the *Steps* metrics, our method consistently requires fewer steps than the ReAct baseline, indicating more efficient task execution. Moreover, our *Replan* number is comparable to ReAct and lower than other communicative baselines, suggesting that despite introducing communication as an additional action, our framework maintains

effective over replanning.

For LLaMA-3.1-8B, although our method slightly underperforms the ReAct baseline, this gap is primarily attributable to the limited ability of this model to effectively utilize the `Talk` action. This limitation is consistent across all communicative baselines on this backbone, including RoCo, CoELA, and CommPARTNR, all of which underperform ReAct. Importantly, our method significantly outperforms all other communicative baselines on LLaMA-3.1-8B, demonstrating its robustness even when the underlying model has weaker communication capabilities.

## 4.5. Results on TDW-MAT

Table 2 reports results on the TDW-MAT benchmark, with performance evaluated separately on the *Food* and *Stuff* object categories. Similar to the results on PARTNR-Dialog, our method consistently outperforms all communicative baselines across models of different sizes and architectures, demonstrating strong robustness and generality.

For the relatively weaker LLaMA-3.1-8B backbone on this task, our method enables substantial self-improvement, surpassing all baseline methods built upon the Gemma-3-12B backbone. For the stronger LLaMA-3.1-70B model, our framework further improves performance on top of an already strong baseline, indicating that the proposed approach remains effective even in high-performance regimes.

*Table 3.* Ablation study on **PARTNR-Dialog** using **Qwen-3-14B**. We analyze the effects of supervised fine-tuning (SFT), stronger inference-only retrieval, and law induction strategies during training and inference.

| Variant | Comp. | Succ. |
|---|---|---|
| **LLawCo (Full)** | **0.84** | **0.74** |
| *w/o SFT* | 0.82 | 0.69 |
| *w/o SFT + RAG* | 0.82 | 0.70 |
| *w/o Laws* | 0.82 | 0.67 |
| *w Manual Laws* | **0.84** | 0.73 |
| *+ Manual Laws* | 0.82 | 0.70 |
| *w/o Filter* | **0.84** | 0.71 |
| *Manual Laws Inf.* | 0.81 | 0.70 |

*Table 4.* Effect of the number of induced laws on **PARTNR-Dialog** (Qwen-3-14B). We vary the maximum number of laws $K$ used by our law-guided framework and report task performance.

| # Laws ($K$) | Comp. | Succ. |
|---|---|---|
| 3 Laws | 0.81 | 0.68 |
| 5 Laws | **0.84** | **0.74** |
| 10 Laws | 0.80 | 0.67 |

*Table 5.* Stability across three complete LLawCo runs with different random seeds on PARTNR-Dialog using Qwen-3-14B.

| Seed | Comp. | Succ. |
|---|---|---|
| 1 | 0.84 | 0.74 |
| 2 | 0.83 | 0.72 |
| 3 | 0.84 | 0.74 |
| Std. | 0.006 | 0.012 |

Most notably, on the Qwen-3-14B backbone, our method achieves particularly significant performance gains, outperforming all baselines based on LLaMA-70B. This result highlights the effectiveness of our law-guided framework in enabling mid-sized models to achieve performance comparable to or exceeding that of substantially larger models.

### 4.6. Ablation Study

We conduct extensive ablation experiments, including both pipeline-level ablations and ablations on the number of induced laws. All ablation studies are conducted using Qwen-3-14B as the backbone LLM to ensure consistency.

**Pipeline Ablations.** Table 3 presents an ablation study analyzing the contribution of different components of our framework. We evaluate several variants to understand the role of each design choice. Removing supervised fine-tuning (*w/o SFT*) leads to a noticeable 5% drop in success rate, highlighting the importance of learning from law-guided training data. To test whether a stronger inference-only strategy can close this gap, we add *w/o SFT + RAG*, which retrieves the most relevant planning trace from the training set and includes it in the prompt at test time, following the ReAct-RAG design in PARTNR (Chang et al., 2024). This variant provides only a limited gain over *w/o SFT*, suggesting that the benefit of LLawCo comes not only from seeing laws or examples at inference time, but from learning successful, law-aligned behaviors through parameter updates. The variant without laws (*w/o Laws*) also shows substantial degradation in success rate, demonstrating that law-guided reasoning is critical for effective cooperation.

We further ablate the data filtering step by training on all episodes without enforcing success or law alignment (*w/o Filter*). This variant results in a clear performance decline, indicating that including failed or law-misaligned episodes introduces noisy or conflicting supervision that negatively impacts learning. We also examine the role of manually specified laws. Replacing automatically induced laws with

manually designed ones (*w/ Manual Laws*), merging induced laws with manual laws (*+ Manual Laws*), or applying manual laws only during inference (*Manual Laws Inf.*) all lead to slight performance drops. These results suggest that laws induced from failure traces capture task and partner specific regularities more effectively than generic human prior knowledge. At the same time, these ablations demonstrate that our framework allows laws to be modified either before or after fine-tuning, enabling controllable behavior changes while largely preserving task performance. Appendix D shows that the same trend holds on Gemma-3-12B.

**Number of Laws.** Table 4 analyzes the effect of varying the number of induced laws. We observe that different choices of the number of laws lead to performance fluctuations. Nevertheless, in most cases, our method consistently outperforms all communicative baseline methods. These results indicate that selecting five laws strikes a favorable balance between expressiveness and stability, and further demonstrate the robustness of our law-guided framework.

### 4.7. Additional Analysis

To assess the stability of the induced laws, we repeat the complete LLawCo pipeline three times with different random seeds on PARTNR-Dialog using Qwen-3-14B. Although the exact wording of the extracted laws varies across runs (see Appendix B), downstream performance remains stable, as shown in Table 5. The induced law sets consistently cover planning, communication or coordination, failure handling, state verification, and task completion, suggesting that the law induction process recovers similar high-level cooperation principles across runs.

We further analyze whether law-grounded reasoning influences action selection. In a controlled TDW-MAT intervention, replacing the law *Begin with a Plan* with the contra-

*Table 6.* Controlled law intervention on TDW-MAT. We report the percentage of 100 random queries for which the model outputs `send message`.

| Setting | Send Message (%) |
|---|---|
| Baseline (No Law) | 55 |
| LLawCo | **92** |
| Intervention | 19 |

*Table 7.* Pairwise first-step action agreement across three law-induction seeds on PARTNR-Dialog.

| Seed Pair | Agreement (%) |
|---|---|
| Seed 1 vs. Seed 2 | 88.4 |
| Seed 1 vs. Seed 3 | 86.4 |
| Seed 2 vs. Seed 3 | 87.2 |

dictory law *Prioritize Immediate Action* changes both the generated thought and the selected action from communication to exploration. As shown in Table 6, across 100 random queries from the same template, the provided laws systematically affect the probability of choosing `send message`. We include a detailed case in Appendix C. Different induced laws also correspond to distinct local action tendencies; for example, *Use Clear and Effective Communication* most often leads to `Talk` actions, while *Use Correct APIs for Tasks* mostly leads to object-manipulation actions. At the same time, first-step action agreement remains high across law-induction seeds (Table 7), indicating that law variation changes local action preferences without destabilizing the overall policy.

### 4.8. Case Study

Our method enables agents to adhere to learned laws during both communication and planning. Figure 3 illustrates a

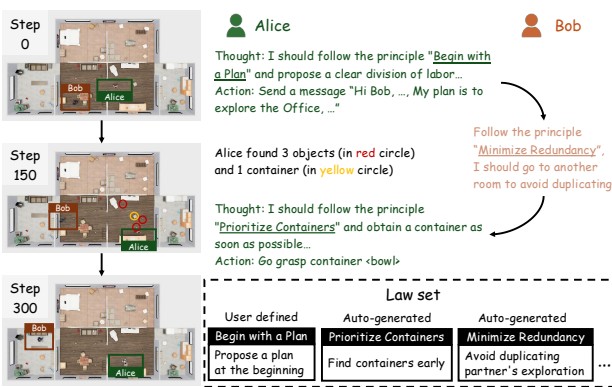

*Figure 3.* **Case study on law-guided inference in TDW-MAT.** During task execution, agents explicitly follow both automatically generated laws and user-defined laws to guide communication and planning, thereby achieving more efficient coordination.

representative example. At Step 0, Alice follows the law *Begin with a Plan* and proposes an explicit exploration plan to Bob. Upon receiving the plan, Bob follows the law *Minimize Redundancy* and chooses to explore a different room to avoid duplicating Alice's efforts. As the episode progresses, Alice discovers three objects and one container; guided by the law *Prioritize Containers*, she decides to first obtain the container. This law-guided decision-making leads to more efficient coordination and significantly improves overall task efficiency.

By following such laws, agents achieve higher task efficiency and more effective cooperation. Moreover, when laws are explicitly specified by the user, the agent reliably adheres to them during task execution, enabling human-defined preferences or constraints to be enforced throughout the cooperative process. For example, in this case study, the law *Begin with a Plan* is user-defined, and the planner consistently follows this law to guide its communication and subsequent decision-making.

## 5. Conclusion

We propose LLawCo, a law-guided learning framework that enables communicative embodied agents to autonomously summarize task and partner specific regularities into laws, learn from them, and explicitly use these laws for reasoning during cooperation. By embedding laws into the agent's reasoning process, our method provides a controllable and interpretable mechanism for aligning agent behavior, while also allowing agents to follow human-specified rules. Across PARTNR-Dialog and TDW-MAT, and four LLM backbones, our approach consistently achieves the best performance among communicative agent methods. The results demonstrate that law-guided learning is an effective and scalable paradigm for improving cooperative embodied agents.

## 6. Limitations

Our experiments focus on two-agent embodied collaboration, following the dominant setting in existing benchmarks such as PARTNR, TDW-MAT, C-WAH, and DialFRED. The LLawCo pipeline itself does not depend on a fixed number of agents, since law extraction, law-guided reflection, and SFT operate over interaction traces; however, larger teams may introduce longer dialog histories, complex credit assignment, and additional context-management challenges. Our method also requires extra training cost for collecting traces, inducing laws, generating law-grounded supervision, and fine-tuning the backbone model. Finally, although explicit laws improve interpretability and make manual inspection possible, they do not by themselves guarantee safety or desirability in deployment, since laws induced from task success can still encode behaviors that require human validation.

## Impact Statement

This paper presents work whose goal is to advance the field of Machine Learning. Our approach may contribute to more interpretable and controllable cooperative embodied agents by representing recurring coordination behaviors as explicit natural-language laws. However, explicit laws should not be viewed as a complete safety mechanism. Because laws are induced from patterns correlated with task success, they may still encode undesirable coordination behaviors in deployment settings. While the law representation makes inspection and manual editing easier, such oversight requires human effort, and realistic applications would require additional validation, safety-aware law induction, or deployment-specific constraints. We do not anticipate direct negative societal impacts from the benchmark and experimental study presented in this paper, but embodied-agent deployment should be evaluated carefully before use in real-world environments.

## Acknowledgement

Chuang Gan and Qinhong Zhou were partially supported by NSF IIS-2441250 and NSF IIS-2404386. Qinhong Zhou conducted part of this work during an internship at MERL and continued the project with partial support from MERL.

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

# A. Law Examples Across Different LLM Backbones

To better understand how different backbone LLMs summarize task and partner specific regularities, we present representative law sets automatically extracted by our framework across multiple LLM backbones. Concretely, for each backbone, we run the agent on the training split to collect its own interaction traces, and then apply our failure-driven law extraction pipeline to summarize these traces into a law set. As a result, the induced laws reflect both the shared structure of the tasks and backbone-specific behaviors and failure modes.

**Qwen-3-14B.**

---

**Plan Before Acting**: Before taking any action, ensure coordination with your partner to avoid redundant actions.
**Use Clear and Effective Communication**: Communicate only when necessary, and ensure that your messages clearly convey your intent and current status. Avoid passive waiting.
**Avoid Infinite Loops**: If an action repeatedly fails, identify the root cause and avoid repeating the same sequence of actions.
**Transport Independently**: Transport objects independently. You cannot transport objects together with your partner, or deliver objects to your partner, or let your partner deliver objects to you.
**Confirm Completion**: Before declaring the task done, double-check that all objects have been placed in the correct locations and that no steps were missed.

---

**Gemma-3-12B.**

---

**Prioritize Proximity**: Always verify proximity to an object before attempting to pick it up or place it, and adjust navigation accordingly.
**Coordinate Object Possession**: Establish clear communication and coordination to avoid conflicts and ensure only one agent attempts to manipulate a specific object simultaneously.
**Validate Action Syntax and Constraints**: Carefully review and validate the syntax of actions, especially spatial relations, and ensure they align with the environment's capabilities.
**Employ Adaptive Replanning**: Continuously monitor the task state and adapt the plan in response to failures, unexpected events, or changes in object availability.
**Manage Dependencies and Prerequisite Actions**: Plan actions sequentially, considering dependencies between them and ensuring prerequisite actions are completed first.

---

**LLaMA-3.1-8B.**

---

**Focus on Task Execution**: Prioritize task execution and avoid unnecessary communication, exploration, or actions unrelated to task completion.
**Communicate Effectively**: Clearly and concisely communicate actions, intentions, and progress to avoid misunderstandings and redundancy.
**Coordinate Actions**: Coordinate actions to ensure agents work toward the same goal without interfering with each other.
**Adapt to Changing Situations**: Adapt plans and strategies to avoid getting stuck in loops or repeating failed actions.
**Plan and Execute Consistently**: Ensure plans align with communications and previous actions to avoid confusion and redundant behavior.

---

**LLaMA-3.1-70B.**

---

**State Consistency**: Maintain an accurate and consistent understanding of the task state, including object locations and status.
**Coordinated Behavior**: Coordinate actions to avoid redundant and conflicting efforts.
**Outcome Verification**: Confirm that tasks have been completed correctly before considering them finished.
**Information Sharing**: Communicate progress, intentions, and status updates to maintain shared situational awareness.
**Dynamic Adaptation**: Adapt plans and actions in response to changes in the task state or unexpected events.

---

**Discussion.**    Despite differences in formulation, several common themes emerge across backbones, including coordination, effective communication, adaptation to failures, and verification of task completion. These shared laws suggest that the failure-driven extraction process consistently captures fundamental principles of cooperative behavior from each backbone's rollout traces.

At the same time, backbone-specific differences reflect varying behaviors and failure patterns during inference. Smaller models (e.g., Gemma-3-12B) tend to induce laws that emphasize low-level execution constraints, such as action syntax validation, proximity checking, and avoiding unnecessary actions, indicating sensitivity to execution and planning errors. In contrast, larger models (e.g., Qwen-3-14B and LLaMA-3.1-70B) more frequently summarize higher-level strategic and coordination principles, such as planning before acting, maintaining state consistency, and verifying outcomes. Overall, these results suggest that law induction is both task-aware and backbone-aware, producing laws that reflect the capabilities and weaknesses of the underlying model.

## B. Stability of Induced Laws

To complement the stability results in the main text, Table 8 reports the induced laws from three complete LLawCo runs with different random seeds on PARTNR-Dialog using Qwen-3-14B.

*Table 8.* Induced laws across random seeds on PARTNR-Dialog using Qwen-3-14B. Similar laws are manually aligned across columns for readability.

| Seed | Law 1 | Law 2 | Law 3 | Law 4 | Law 5 |
|------|-------|-------|-------|-------|-------|
| 1 | Plan Before Acting | Use Clear and Effective Communication | Avoid Infinite Loops | Transport Independently | Confirm Completion |
| 2 | Plan with Shared Goals | Coordinate Actions | Handle Failures Proactively | Verify Object States | Track Task Progress |
| 3 | Plan Actions Strategically | Track and Communicate Task Completion | Avoid Unnecessary Waiting | Use Correct APIs for Tasks | Verify Object State |

## C. Law-Action Analysis

We analyze whether different induced laws correspond to distinct action tendencies. For example, *Use Clear and Effective Communication* leads to `Talk` actions in 91.3% of the cases, while *Plan Before Acting* leads to `Navigate` actions in 48.5% of the cases. The law *Use Correct APIs for Tasks* leads to object-manipulation actions, such as `Clean`, `Fill`, `PowerOn`, `Open`, and `TurnOn`, in 91.9% of the cases. These results suggest that the laws are not merely interchangeable text strings, but tend to ground different local action choices.

At the same time, the overall first-step policy remains stable across seeds. We compare first-step actions on the PARTNR-Dialog validation set because later actions depend on divergent histories and are therefore less directly comparable.

We also conduct a controlled intervention experiment on TDW-MAT to test whether changing a law changes the generated reasoning and action. As shown in Table 9, replacing *Begin with a Plan* with *Prioritize Immediate Action* changes both the thought and the selected action under the same state.

*Table 9.* Controlled law intervention under the same TDW-MAT state.

| Setting | Input Prompt | Thought | Action |
|---------|-------------|---------|--------|
| Baseline | Same state; no laws | Since I have not found the bed yet, exploring might help locate it, but the immediate priority is gathering objects. Therefore, the best action is to explore the current room. | explore current room |
| LLawCo | Same state; full laws | According to the *Begin with a Plan* principle, the agent should propose a clear plan. The message informs the partner of the current plan and suggests complementary exploration. | send a message |
| Intervention | *Begin with a Plan* replaced by *Prioritize Immediate Action* | The agent should prioritize immediate exploration to locate useful containers and avoid delaying physical progress. | explore current room |

## D. Additional Ablations

Table 10 reports a representative subset of ablations on Gemma-3-12B. The trend is consistent with the Qwen-3-14B ablation in the main text: removing SFT, removing laws, or using manual laws only at inference time decreases performance.

*Table 10.* Additional ablation study on PARTNR-Dialog using Gemma-3-12B.

| Variant | Comp. | Succ. |
|---|---|---|
| **LLawCo (Full)** | **0.79** | **0.61** |
| *w/o SFT* | 0.76 | **0.61** |
| *w/o Laws* | 0.77 | 0.59 |
| *Manual Laws Inf.* | 0.76 | 0.59 |

We also compare failure-only law induction with a mixed setting that derives laws from both failed and successful episodes. Table 11 shows that the two variants achieve similar performance, suggesting that both sources can yield useful laws, while failure traces provide a direct way to target recurring coordination errors.

*Table 11.* Comparison of law induction sources on PARTNR-Dialog using Qwen-3-14B.

| Variant | Law Induction Source | Comp. | Succ. |
|---|---|---|---|
| **LLawCo (Full)** | Failure episodes | 0.84 | **0.74** |
| Mixed Laws | Failure + Successful episodes | **0.85** | 0.73 |

*Table 12.* Frequency of each induced law in the reflected training data for Gemma-3-12B.

| Law | Frequency (%) |
|---|---|
| Prioritize Proximity | 30.32 |
| Coordinate Object Possession | 9.57 |
| Validate Action Syntax and Constraints | 7.89 |
| Employ Adaptive Replanning | 4.48 |
| Manage Dependencies and Prerequisite Actions | 47.74 |

## E. Statistical Summary of Induced Laws

We provide additional statistics on the induced laws for Gemma-3-12B. Table 12 reports how frequently each law is used in the reflected training data. Table 13 summarizes the distribution of laws generated across 25 random seeds after grouping them into high-level coordination categories.

## F. Implementation Details of PARTNR-Dialog

### F.1. Backward Compatibility

The PARTNR-Dialog Benchmark maintains full backward compatibility with the original PARTNR benchmark. The communication and deliver functionalities are implemented as optional extensions: agents may operate without using the `Talk` actions, in which case the benchmark behavior degenerates to that of the original PARTNR environment. This design allows researchers to directly compare communication-enabled and communication-free approaches within the same benchmark framework, facilitating controlled ablation studies on the role of communication in multi-agent coordination.

### F.2. Visualization Tools

As shown in Figure 4, building upon PARTNR's existing third-person visualization with action annotations, we extend the visualization system with three key enhancements. First, we add a top-down view video that provides a bird's-eye perspective, enabling clearer analysis of spatial relationships and coordination strategies. Second, we explicitly visualize `Talk` actions by rendering the communication content alongside action annotations. Third, for instantaneous actions includeing `Talk`, the visualization system briefly pauses by extending the display duration through repeating the corresponding frame 50 times, allowing viewers to better observe the communication events and their context.

*Table 13.* Category distribution of induced laws for Gemma-3-12B across 25 random seeds.

| Category | Share |
|---|---|
| Verify Task Progress | 0.23 |
| Adapt Strategies | 0.20 |
| Effective Communication | 0.20 |
| Coordinate Actions | 0.19 |
| Verify Task Completion | 0.18 |

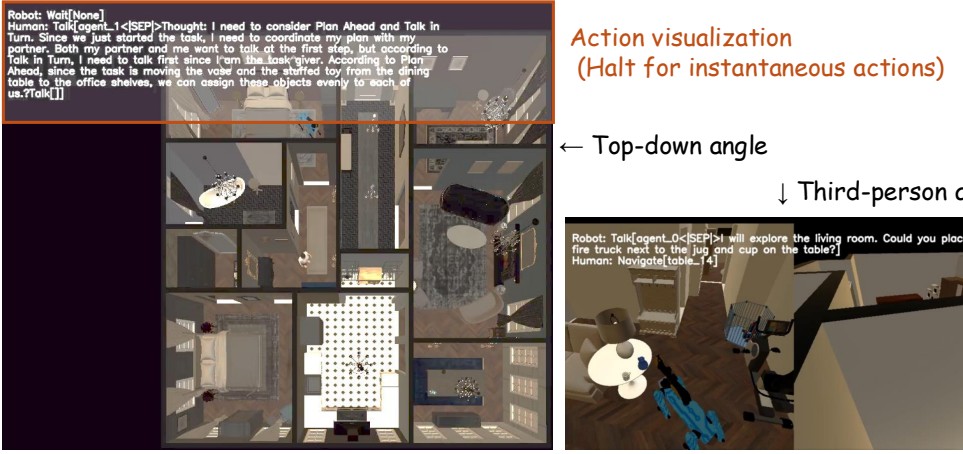

*Figure 4.* **Visualization tools.** We extend the original PARTNR visualization system by adding a top-down view. In addition, for instantaneous actions such as `Talk`, the visualization briefly pauses, allowing clearer inspection.

## G. Case Study on PARTNR-Dialog

Figure 5a presents a qualitative case study on PARTNR-Dialog using the Qwen-3-14B backbone, demonstrating law-guided planning and communication in a human–robot cooperative task. At the early stage of the episode, the robot follows the law *Plan Before Acting* to explore the environment and gather relevant information before committing to task execution. As the task progresses, the human partner leverages the law *Use Clear and Effective Communication* to issue precise and unambiguous instructions, ensuring that intentions and task status are clearly shared. This example illustrates how law-guided inference with a stronger backbone emphasizes high-level planning and coordination, leading to effective cooperation and improved task efficiency on PARTNR-Dialog.

Figure 5b shows a another case study using the Gemma-3-12B backbone. When executing a syntactically complex `Place` action, the agent explicitly invokes the law *Validate Action Syntax and Constraints*, focusing on verifying action arguments and environmental feasibility before execution. This behavior reflects common failure modes observed for this backbone in the training traces, where invalid action formats or constraint violations frequently lead to execution errors. In the final stage of the episode, before issuing the `Done` signal, the agent follows the law *Manage Dependencies and Prerequisite Actions* to review whether all task dependencies have been satisfied and actions have been completed in the correct order. This example highlights how law-guided inference adapts to backbone-specific weaknesses, enabling more reliable execution and robust task completion.

## H. Prompts Used in Experiments

This prompt ($\mathcal{T}_{failure}$) is used for failure analysis:

You are analyzing a failed multi-agent planning task. Below is the trace of the episode:

{trace}

Analyze this trace and identify the main reason why the task failed. Be specific and concise. Focus on what went wrong

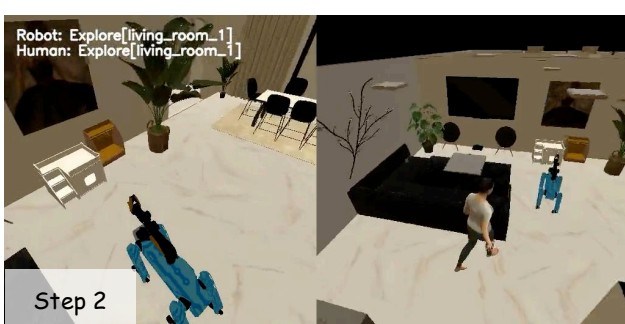
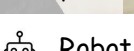
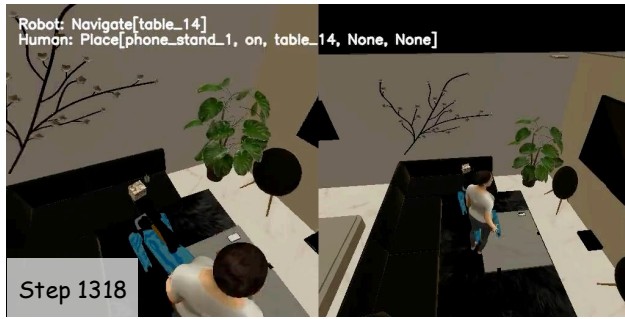

👤 Robot

Thought: According to the "Plan Before Acting" principle, I should begin by exploring the relevant room to gather necessary information before taking further steps.
Action: Explore[living_room_1]

👤 Human

Thought: According to "Use Clear and Effective Communication," it is important to ensure that my message clearly conveys my intent and current status to avoid confusion or redundant actions.
Action: Talk[Please place the candle on the coffee table (table_14) in the living room.]

*(a)* Case study on PARTNR-Dialog with the Qwen-3-14B model.

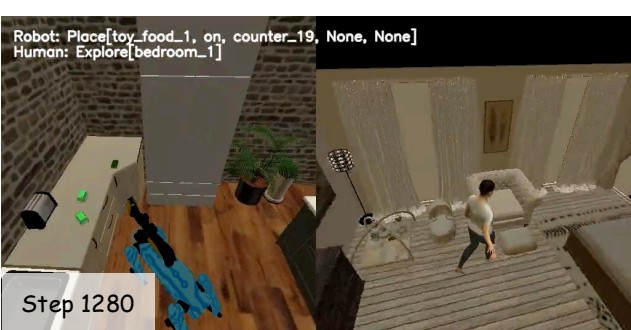
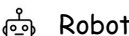
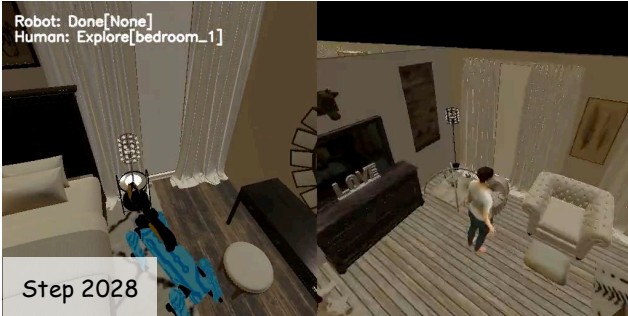
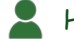

👤 Robot

Thought: I am now next to the counter. I should place the toy food on it, ensuring I adhere to Validate Action Syntax and Constraints to confirm the placement is valid and feasible given the current environment.
Action: Place[toy_food_1, on, counter_19, None, None]

👤 Human

Thought: I've validated that all requirements are met, aligning with Manage Dependencies and Prerequisite Actions to ensure all steps were completed sequentially and correctly.
Action: Done[]

*(b)* Case study on PARTNR-Dialog with the Gemma-3-12B model.

*Figure 5.* Case studies on PARTNR-Dialog demonstrating law-guided planning and communication.

in the planning or execution that led to the failure. Consider:

- Action execution problems

- Task understanding mistakes

- Coordination and communication failures

- Planning errors or inconsistencies

Provide a clear, concise failure reason in 1-3 sentences.

Failure reason:

This prompt ($\mathcal{T}_{law}$) is used for law summary:

You are analyzing multiple failure cases from multi-agent planning tasks. Below are the failure reasons extracted from failed traces:

{failure_reasons}

Based on these failure reasons, identify the key principles that agents should follow to avoid these failures. Generate at most {max_law} laws.

Each principle should be: 1. Clear, concise (one sentence), and actionable 2. General enough to apply to various situations 3. Address common failure patterns observed in the traces

Format your response as a JSON object with law names as keys and descriptions as values. For example: {{ "Law Name 1": "Description of law 1", "Law Name 2": "Description of law 2", "Law Name 3": "Description of law 3", "Law Name 4": "Description of law 4", "Law Name 5": "Description of law 5" }}

Generate the principles now:

This prompt ($\mathcal{T}_{reflect}$) is used for law-guided data generation:

The action {action} has already been chosen. Your task is to regenerate the thought.

Reference the principle that best matches the original thought, using its EXACT name from the principles listed below ({all_principle_name})

Principles to follow (names and content): {principle_description}

Original thought: {original_thought}

Write your regenerated thought starting with "Thought: " and keep it as a single, natural paragraph.

Example ("principle_name" would be replaced by the name of an actual principle):

Regenerated thought: Thought: I need to pick up the cup from the counter and place it on the table to complete the task. According to principle_name, I should focus on task-relevant actions to make progress.

