# OpenReview forum: "LLawCo: Learning Laws of Cooperation for Modeling Embodied Multi-Agent Behavior"
_ICML.cc/2026/Conference — ICML 2026 regular_

### Official Review · Reviewer_96jq · 2026-03-04

**Soundness:** 2
**Presentation:** 3
**Significance:** 3
**Originality:** 3
**Overall Recommendation:** 4
**Confidence:** 3

**Summary:**

This paper proposes LLawCo, an embodied multi-agent collaborative learning framework based on cooperative rules. LLawCo deduces high-level laws from failure trajectories, uses these laws to filter more consistent success trajectories, and generates supervised thinking and action data with law references. Supervisory fine-tuning then ensures that agents consistently follow these rules during planning and communication. The authors also propose PARTNR-Dialog as a scalable dialogue-based benchmark for PARTNR to support messaging and large-scale evaluation. Experimental results show that LLawCo provides a stable improvement over existing communication collaboration baselines.

**Compliance With Llm Reviewing Policy:**

Affirmed.

**Final Justification:**

I will retain my positive assessment and keep my score.

**Key Questions For Authors:**

1. The paper mentions that current training only requires the thinker to cite at least one law, while Treflect tends to choose the best-matching law. The reasoning process also typically only requires citing relevant laws. Please clarify the following points in your rebuttal:
a. Whether citing multiple laws simultaneously is allowed or encouraged at each decision step.
b. How does the model prioritize and decide on the laws when multiple laws apply simultaneously or there is tension?
c. Does the tinference contain any explicit instructions on choosing laws/handling conflicts?

2. Regarding the second weakness, please explain how to avoid this self-confirmation bias. For example, have you observed significant changes in laws under different random seeds or different subsets of failures? Does the model's performance remain stable when laws change? Please provide basic stability analysis to demonstrate that this risk is manageable in practice.

3. Please provide a brief statistical summary of the final distribution and coverage of laws, such as the frequency, repetition rate, or clustering results of laws corresponding to different failure types, and explain which laws contribute the most.

**Limitations:**

see weakness

**Strengths And Weaknesses:**

1.Strengths

a. The authors designed the LLawCo framework, extracting laws from failure examples and then using these laws to filter data for supervised fine-tuning. The overall method is simple, and the training objective is clear: to directly instruct the model to produce thought processes and actions with law references during planning and communication. This framework has application value for customized agent interactions.
b. Using laws is also a sensible design choice, since it introduces an explicit real-world rule layer and makes the system easier to interpret and control. It also helps with diagnosing collaboration failures and applying manual fixes, which is harder with purely black-box strategies.
c. The method is tested with four LLM backbones on two writing benchmarks, and the overall improvements look reasonably consistent.

2.Weaknesses

a. Currently, the paper mainly uses laws as textual constraints in the prompt, lacking a clear mechanism to explain how these natural language rules stably translate into specific Talk and Act decisions. Especially when multiple laws apply simultaneously, how does the model choose which to apply? I believe this needs discussion; otherwise, the narrative remains largely intuitive.
b. I understand the authors' reason for wanting a method that doesn't rely on stronger models or manual annotation. However, in the current process, the summarization of laws, the judgment of law alignment, and the generation of training data are basically all done by models at the same level. This easily leads to the risk of "the model teaching itself with its own preferences," thus solidifying the bias. The paper does not provide sufficient analysis to demonstrate that this risk is controllable in practice.
c. The paper lacks statistical analysis of the overall quality and coverage of laws. Currently, it mainly relies on a few examples, making it difficult for readers to judge whether laws are duplicated, whether they cover the main failure modes, etc.

---

> ### Author Rebuttal · Authors · 2026-03-31
>
> We thank the reviewer for the thoughtful feedback and constructive concerns, which we address below.
>
> > How does LLawCo handle law selection at each decision step, especially when multiple laws apply or conflict, and does $T_{inference}$ provide any explicit rule for this?
>
> We thank the reviewer for these clarifying questions.
>
> 1. In our current implementation, the model is trained to ground each decision in a single most relevant law. This design simplifies the reasoning process and encourages focused law grounding.
> 2. When multiple laws could apply, the current reflection prompt explicitly instructs the model to identify the single most relevant law for the current decision context. As a result, our present implementation uses best-match selection by LLM rather than multi-law composition or explicit conflict resolution.
> 3. The inference template ($T_{inference}$) provides the full set of laws and instructs the model to follow them, but does not explicitly specify how to select among them or handle conflicts. Instead, this behavior is learned implicitly through training. We also observe in controlled intervention examples (by modifying the input laws, with Qwen-14B backbone on TDW-MAT) that both the generated thoughts and the resulting actions change consistently, further explains how these natural language rules stably translate into specific Talk and Act decisions. We will add this study in our paper.
>
> | Setting | Input Prompt | Thought | Action |
> |-|-|-|-|
> | Baseline | no laws | ... Additionally, since I haven't found the bed yet, exploring might help locate it, but the immediate priority is gathering objects. Therefore, the best action is to explore the current room. | explore current room |
> | LLawCo | full laws | First, according to the "Begin with a Plan" principle, she should propose a clear plan. The message in option A does that by informing Bob of her actions and suggesting he explore other rooms ... | send a message |
> | Intervention | *Begin with a Plan* → *Prioritize Immediate Action* | ... Prioritize Containers by exploring the current room to locate a container, enabling efficient transport of multiple objects later, aligning with the principle to avoid redundant exploration and act independently. | explore current room |
>
> > How sensitive is LLawCo to self-confirmation bias, and do the induced laws and downstream performance remain stable across different random seeds or subsets of failures?
>
> We thank the reviewer for this helpful suggestion. Evaluating stability across random seeds requires re-running the full pipeline, including law extraction, SFT, and inference, which is computationally expensive. Due to limited time, we repeat the complete LLawCo pipeline three times with different random seeds for one representative setting (Qwen-3-14B on PARTNR-Dialog) and report both the extracted laws and the corresponding performance. The results indicate that, while the exact wording of the laws varies across runs, the downstream performance is stable (we manually reorder the laws in each row so that similar laws appear in aligned columns). We will add this result in our paper.
>
> | Seed | Law 1 | Law 2 | Law 3 | Law 4 | Law 5 | Comp. | Succ. |
> |-|-|-|-|-|-|-:|-:|
> | 1 | Plan Before Acting | Use Clear and Effective Communication | Avoid Infinite Loops | Transport Independently | Confirm Completion | 0.84 | 0.74 |
> | 2 | Plan with Shared Goals | Coordinate Actions | Handle Failures Proactively | Verify Object States | Track Task Progress | 0.83 | 0.72 |
> | 3 | Plan Actions Strategically | Track and Communicate Task Completion | Avoid Unnecessary Waiting | Use Correct APIs for Tasks | Verify Object State | 0.84 | 0.74 |
> | Mean ± Std | — | — | — | — | — | 0.006 | 0.012 |
>
> > Can the authors provide a brief statistical summary of the induced laws, including their distribution, repetition patterns, clustering by failure type, and relative contribution?
>
> We thank the reviewer for this helpful suggestion. To address this, we add two analyses. First, we report the frequency of each law in the reflected training data. Second, we analyze the distribution of induced laws by repeatedly generating laws under 25 random seeds and grouping them into high-level coordination categories, which helps assess whether the learned laws are duplicated or concentrated around a small number of failure patterns. We will add these analyses in our paper.
>
> * Law frequency (Gemma-3-12B)
>
> | Law | Frequency (%) |
> |-|-:|
> | Prioritize Proximity | 30.32 |
> | Coordinate Object Possession | 9.57 |
> | Validate Action Syntax and Constraints | 7.89 |
> | Employ Adaptive Replanning | 4.48 |
> | Manage Dependencies and Prerequisite Actions | 47.74 |
>
> * Law category distribution (Gemma-3-12B)
>
> | Category | Share |
> |-|-:|
> | Verify Task Progress | 0.23 |
> | Adapt Strategies | 0.20 |
> | Effective Communication | 0.20 |
> | Coordinate Actions | 0.19 |
> | Verify Task Completion | 0.18 |

---

> > ### Author Rebuttal · Reviewer_96jq · 2026-04-02
> >
> > Thank the author for their response. I will retain my positive assessment and keep my score.

---

> > > ### Author Response · Authors · 2026-04-03
> > >
> > > We sincerely thank the reviewer for the comments and keeping the positive assessment. We are glad that our responses addressed your concerns.

---

### Official Review · Reviewer_jQKf · 2026-03-09

**Soundness:** 2
**Presentation:** 3
**Significance:** 3
**Originality:** 3
**Overall Recommendation:** 5
**Confidence:** 4

**Summary:**

LLawCo addresses the challenge of aligning LLM-based embodied agents with both their cooperation partners and task environments by autonomously deriving behavioral laws from interaction data, without relying on stronger models or human annotation. The framework operates in three stages: failed training episodes are analyzed to extract recurring coordination breakdowns, which are turned into laws; successful episodes are then filtered for law-alignment and used to generate law-referencing reasoning traces; finally, these traces serve as supervised fine-tuning data. The learned laws are also added to the prompt at inference time. Alongside the method, the authors introduce PARTNR-Dialog, a large-scale communicative cooperation benchmark extending PARTNR. LLawCo consistently outperforms on both PARTNR-Dialog and TDW-MAT.

**Compliance With Llm Reviewing Policy:**

Affirmed.

**Final Justification:**

I thank the authors for their thoughtful response to my questions and have raised my score to a 5.

**Key Questions For Authors:**

1. The paper reports a single set of induced laws per experimental condition. How stable are the extracted laws across different runs of the law extraction pipeline? Specifically, do different random seeds produce qualitatively similar laws, or does the law set vary substantially? If the latter, how sensitive are the downstream performance gains to the particular laws that happen to be extracted?

2. The framework's interpretability claims rest on the assumption that law-referencing thoughts are causally connected to the actions selected. What evidence do the authors have that this is the case, rather than the thoughts being post-hoc rationalizations? For instance, does intervening on the thought, i.e. providing a thought that references a different or contradictory law, systematically change the action output of the fine-tuned model?

3. Can the authors provide examples of laws that were extracted? More broadly, what prevents the framework from inducing laws that are performance-improving but encode coordination behaviors that would be considered unsafe or undesirable in deployment?

**Limitations:**

No, as noted in weaknesses: the impact statement should engage with directly with the possibility of learning undesirable behaviors rather than treating interpretability as a sufficient safeguard.

**Strengths And Weaknesses:**

**Strengths**
- **Ablation study.** The ablation study is thorough and well-structured. Testing both pipeline-level components (removing SFT, removing laws, removing filtering) and the law count hyperparameter gives a clear picture of what each design choice actually contributes. The finding that automatically induced laws outperform manually specified ones is a particularly interesting result that is directly supported by the ablation evidence.
- **Self-improvement without stronger models.** Most prior learning-based approaches in this space distill from stronger models; the fact that LLawCo improves purely through self-alignment is a meaningful contribution.
- **Benchmark.** In addition to contributing a new method for alignment, the authors also contribute a training set for communicative embodied agents, which seems like a useful community contribution independent of the method itself.

**Weaknesses**
- **The learned laws may encode undesirable behaviors.** The framework validates laws by their correlation with task success, with no mechanism to assess whether the learned principles are ethically sound or otherwise desirable. A law that boosts performance metrics could simultaneously encode problematic coordination behaviors, like ignoring a partner's requests or acting unilaterally, and the framework would have no way to flag this. The authors present human editability of laws as a controllability and safety feature, but interpretability alone does not guarantee that what is being learned is beneficial; auditing laws for subtle misalignment at scale would require significant human oversight that the paper does not discuss. The impact statement should engage with this directly rather than treating interpretability as a sufficient safeguard.
- **Faithfulness of reasoning traces.** The framework assumes that the law-referencing thoughts generated by the backbone LLM faithfully reflect the reasoning behind the selected actions. However, it is well established that LLM-generated reasoning traces can be post-hoc rationalizations rather than genuine explanations. If the thoughts are not causally connected to the actions, the interpretability and controllability claims of the framework are significantly weakened.
- **Underdeveloped case study.** The case study in Figure 3 presents a single, seemingly favorable example of law-guided behavior, but provides no indication of whether it is representative or cherry-picked. More critically, the paper offers little systematic analysis of the laws that are actually learned across different runs, environments, or backbones. Understanding whether the extracted laws are consistent, interpretable, and genuinely novel would substantially strengthen the paper's qualitative contributions and its controllability claims.
- **Single-run results with no variance reported.** The paper reports point estimates throughout with no standard deviations or confidence intervals across seeds. Given that law extraction involves stochastic LLM calls and fine-tuning, it is unclear how stable the results are. The ablation on K hints at sensitivity, but the variance question is not addressed directly.

Note: there are some minor typos, like in Section 3.5, which references 6 stages but only lists 5.

---

> ### Author Rebuttal · Authors · 2026-03-31
>
> We thank the reviewer for the thoughtful feedback and constructive concerns, which we address below.
>
> > How stable are the induced laws across different runs, and how sensitive is downstream performance to variation in the extracted law set?
>
> We thank the reviewer for this important question. Evaluating stability across random seeds requires re-running the full pipeline, including law extraction, SFT, and inference, which is computationally expensive. Due to limited time, we repeat the complete LLawCo pipeline three times with different random seeds for one representative setting (Qwen-3-14B on PARTNR-Dialog) and report both the extracted laws and the corresponding performance. The results indicate that, while the exact wording of the laws varies across runs, the downstream performance is stable (we manually reorder the laws in each row so that similar laws appear in aligned columns). We will add this result in our paper.
>
> | Seed | Law 1 | Law 2 | Law 3 | Law 4 | Law 5 | Comp. | Succ. |
> |-|-|-|-|-|-|-:|-:|
> | 1 | Plan Before Acting | Use Clear and Effective Communication | Avoid Infinite Loops | Transport Independently | Confirm Completion | 0.84 | 0.74 |
> | 2 | Plan with Shared Goals | Coordinate Actions | Handle Failures Proactively | Verify Object States | Track Task Progress | 0.83 | 0.72 |
> | 3 | Plan Actions Strategically | Track and Communicate Task Completion | Avoid Unnecessary Waiting | Use Correct APIs for Tasks | Verify Object State | 0.84 | 0.74 |
> | Mean ± Std | — | — | — | — | — | 0.006 | 0.012 |
>
> > Is law-grounded reasoning truly influencing action selection, or are the generated thoughts merely post-hoc rationalizations?
>
> We thank the reviewer for this insightful question. In our framework, thoughts are generated autoregressively before actions and are trained jointly with them, so they are not produced purely as post-hoc explanations.
>
> To more directly examine this, we conduct a controlled intervention experiment with Qwen-14B backbone on TDW-MAT. Specifically, we design three settings: (1) a baseline without laws, (2) our standard law-guided model, and (3) an intervention setting with a contradictory law. The table below summarizes how the same state maps to different actions under these inputs. We will add this study in our paper.
>
> | Setting | Input Prompt | Thought | Action |
> |-|-|-|-|
> | Baseline (no law) | Same state; no laws | ... Additionally, since I haven't found the bed yet, exploring might help locate it, but the immediate priority is gathering objects. Therefore, the best action is to explore the current room. | explore current room |
> | LLawCo (with law) | Same state; full laws | First, according to the "Begin with a Plan" principle, she should propose a clear plan. The message in option A does that by informing Bob of her actions and suggesting he explore other rooms ... | send a message |
> | Intervention (contradictory law) | Same state; *Begin with a Plan* → *Prioritize Immediate Action* | ... Prioritize Containers by exploring the current room to locate a container, enabling efficient transport of multiple objects later, aligning with the principle to avoid redundant exploration and act independently. | explore current room |
>
> > Can the authors provide example extracted laws, and how does the framework avoid inducing laws that improve performance but encode unsafe or undesirable coordination behaviors?
>
> We thank the reviewer for this thoughtful comment. We provided representative examples of extracted laws in Appendix A. We agree that better task performance does not guarantee that the induced laws are always desirable. In particular, laws correlated with success could still encode undesired coordination behaviors, and our current framework does not explicitly rule this out.
>
> We also agree that interpretability and editability help, but are not sufficient safeguards. We will revise the paper to state this limitation more clearly and expand the impact statement accordingly.
>
> * Planned addition to Impact Statement:
> "Although our framework represents coordination behavior as explicit laws, this does not guarantee that the learned laws are always beneficial or aligned with broader safety considerations. Because laws are induced based on their correlation with task success, the system may learn behaviors that improve performance while still being undesirable in deployment. While explicit laws make inspection and manual editing possible, such modification requires human effort. We therefore view interpretability as a useful feature, but not as a sufficient safeguard, and additional validation or safety-aware mechanisms are needed in realistic deployment settings."
>
> > Typos
>
> Thank you and we will fix them.

---

> > ### Author Rebuttal · Reviewer_jQKf · 2026-04-02
> >
> > Thank you for your rebuttal!
> >
> > Regarding Q1, this is actually an interesting finding and should be further investigated. It seems like the difference across seeds goes beyond the "exact wording" of laws -- for example, "Use Correct API for Tasks" from the Seed 3 run doesn't seem to have an analogue in the Seed 1 run. Do these different laws translate to the same actions?
> >
> > Regarding Q2, thanks for the clarification and additional experiment. What is your sample size? What metrics will you report?
> >
> > Regarding Q3, I apologize for my oversight -- thanks for referring me to your Appendix for example laws. I also appreciate the inclusion of the impact statement to address my concern.
> >
> > I am inclined to raise my score but would appreciate clarification about the Q1 and Q2 experiments.

---

> > > ### Author Response · Authors · 2026-04-03
> > >
> > > > Regarding Q1, this is actually an interesting finding and should be further investigated. It seems like the difference across seeds goes beyond the "exact wording" of laws -- for example, "Use Correct API for Tasks" from the Seed 3 run doesn't seem to have an analogue in the Seed 1 run. Do these different laws translate to the same actions?
> > >
> > > We thank the reviewer for this helpful follow-up. This is a subtle but important observation: some seed differences go beyond wording. To better understand whether these differences affect actions, we analyze the action distribution associated with each law and observe several clear patterns. For example, `Use Clear and Effective Communication` leads to *Talk* actions in **91.3%** of the cases, and `Plan Before Acting` leads to *Navigate* actions in **48.5%** of the cases. For the law you specifically pointed out, `Use Correct APIs for Tasks`, **91.9%** of the output actions are object-manipulation actions, including *Clean*, *Fill*, *PowerOn*, *Open*, *TurnOn*, and etc. These APIs are typically more complex (often involve strict object- or agent-specific constraints).
> > >
> > > At the same time, to analyze the overall behavioral effect across seeds, we compare the first-step actions on the PARTNR-Dialog validation set, since later actions depend on different histories and are therefore harder to compare directly. The pairwise action agreement rates are:
> > >
> > > | Pair | % Agreement |
> > > |---|---:|
> > > | Seed 1 vs Seed 2 | 88.4 |
> > > | Seed 1 vs Seed 3 | 86.4 |
> > > | Seed 2 vs Seed 3 | 87.2 |
> > >
> > > These results suggest that different laws can indeed lead to different local actions, but without clear semantic conflicts, the resulting behaviors remain highly similar at the overall policy level. Although Seed 3 is slightly less aligned with Seeds 1 and 2, the majority of first-step actions are still consistent across seeds.
> > >
> > > We will summarize these law–action relations in a figure and a table and include this analysis in the revised paper.
> > >
> > > > Regarding Q2, thanks for the clarification and additional experiment. What is your sample size? What metrics will you report?
> > >
> > > Thank you for asking to detail this point better. We concur that a quantitative analysis can make our study much stronger. To this end, we constructed 100 random queries from the same input template and measured the percentage of cases in which the model outputs the *send message* action under the three settings we described in our response above:
> > >
> > > | Setting | % Send Message |
> > > |---|---:|
> > > | Baseline (No Law) | 55 |
> > > | LLawCo | 92 |
> > > | Intervention | 19 |
> > >
> > > The result provides additional evidence that when incorporating an intervention that  contradicts a selected law, the input systematically changes the action distribution: under *LLawCo*, the model is much more likely to choose `send message` (92%), while under *Intervention*, it is significantly less (19%).

---

### Official Review · Reviewer_M6RH · 2026-03-12

**Soundness:** 3
**Presentation:** 3
**Significance:** 3
**Originality:** 3
**Overall Recommendation:** 5
**Confidence:** 3

**Summary:**

The paper proposes LLawCo, a cooperative learning framework that extracts behavioral laws from failures and leverages law-consistent successful trajectories for reasoning-based supervised fine-tuning. It also introduces the PARTNR-Dialog benchmark for evaluating multi-agent communication and cooperative planning.

**Compliance With Llm Reviewing Policy:**

Affirmed.

**Final Justification:**

The supplementary content added by the author dispelled my doubts.

**Key Questions For Authors:**

see above.

**Limitations:**

yes

**Strengths And Weaknesses:**

**Strengths**
1. This paper proposes LLawCo, a cooperative learning framework that extracts behavioral laws from failed episodes and integrates them into agents’ reasoning via supervised fine-tuning. This failure-driven mechanism helps mitigate behavioral misalignment and task inconsistency in decentralized, partially observable environments. Notably, the approach enables agent self-alignment without relying on stronger teacher models or human annotations.
2. Comprehensive experiments across multiple LLM backbones (LLaMA-3.1, Gemma3, Qwen-3) and benchmarks (PARTNR-Dialog, TDW-MAT) demonstrate consistent improvements over strong communicative-agent baselines. Ablation studies further validate the contributions of key components.
3. The paper introduces PARTNR-Dialog, a large-scale benchmark for embodied multi-agent communication and cooperation. By extending the PARTNR environment with communication infrastructure and scalable evaluation support, it provides a useful platform for future research in multi-agent collaboration.
4. The paper is clearly written and well-structured. The methodology, experimental setup, and results are presented in a logical and coherent manner, making the overall contribution easy to follow.

**Weaknesses**
1. The framework derives cooperative laws solely from failed episodes and filters successful trajectories based on these laws. This failure-centric design may bias the learned rules toward error avoidance while overlooking positive cooperation patterns present in successful interactions, potentially limiting the richness and generalization of the learned behaviors.
2. Ablations are conducted only on the Qwen-3-14B backbone. It remains unclear whether the observed component contributions generalize to other LLM backbones used in the experiments; validating on at least one additional model would strengthen the claims.

---

> ### Author Rebuttal · Authors · 2026-03-31
>
> We appreciate the reviewer’s thoughtful comments and helpful questions, and respond to them below.
>
> > Could deriving laws only from failures bias LLawCo toward error avoidance and miss positive cooperation patterns from successful interactions?
>
> We thank the reviewer for this insightful observation. We would like to clarify that LLawCo is not purely failure-driven at the behavior level. While laws are induced from failed episodes, the final training data is constructed from successful, law-aligned trajectories, which ensures that the learned policy is grounded in effective cooperation rather than failures.
>
> We also agree that deriving laws solely from failed episodes may bias them toward error avoidance. To better evaluate this, we will extend our experiments with a mixed law induction setting, where laws are derived from both failure traces (capturing error patterns) and successful traces (capturing effective cooperation strategies):
>
> | Variant | Law Induction Source | Comp. | Succ. |
> |-|-|-:|-:|
> | Full LLawCo | Failure episodes | 0.84 | 0.74 |
> | Mixed Laws | Failure + Successful episodes | 0.85 | 0.73 |
>
> The two variants achieve very similar performance, suggesting that both failure-driven and mixed law induction are valid choices in LLawCo. We will include this analysis in the revision to better understand the role of failure-driven versus success-driven law induction. We will add this result and part of discussion in our paper.
>
> > Do the ablation results generalize beyond Qwen-3-14B, and can they be validated on at least one additional backbone?
>
> We thank the reviewer for this helpful suggestion. We agree that validating the ablations on more than one backbone would strengthen the claim. Due to time constraints, we run a representative subset of the ablations on an additional backbone (Gemma-3-12B) and include these results in the revision. The overall trend is consistent with the Qwen-3-14B results: removing SFT, removing laws, or replacing them with inference-only manual laws leads to a clear drop.
> | | Comp. | Succ. |
> |-|-:|-:|
> | Full LLawCo | 0.79 | 0.61 |
> | w/o SFT | 0.76 | 0.61 |
> | w/o Laws | 0.77 | 0.59 |
> | Manual Laws Inf. | 0.76 | 0.59 |

---

> > ### Author Rebuttal · Reviewer_M6RH · 2026-04-01
> >
> > Thank you for the authors’ detailed response. I will raise my score to 5.

---

> > > ### Author Response · Authors · 2026-04-01
> > >
> > > We sincerely thank the reviewer for the feedback and raising the score. We are glad that our responses addressed your concerns.

---

### Official Review · Reviewer_pkxZ · 2026-03-13

**Soundness:** 3
**Presentation:** 3
**Significance:** 2
**Originality:** 3
**Overall Recommendation:** 3
**Confidence:** 3

**Summary:**

The authors identify a problem in the current state of multi-agent LLM architectures, in that existing collaborations are often misaligned in terms of learning behavior conditioned on both team-state and task-state. They propose a framework called LLawCo, to enable agents utilize high-level laws to align individual actions with the targeted collective behavior. To enable this, the authors design an approach to fine-tune a Law-aligned LLM, wherein the llm is trained to generate think-and-act traces based on the current observation history, and laws. Furthermore, the authors expand the PARTNR benchmark, to add communication as well as a training split for fine-tuning, called PARTNR-Dialog. LlawCo is compared to four other baselines in PARTNR-Dialog and TDW-MAT, to highlight the benefits of law-aligned action generation in multi-agent settings. Finally, the authors conduct an ablation study to analyze the role of each design choice.

**Compliance With Llm Reviewing Policy:**

Affirmed.

**Final Justification:**

The rebuttal addressed my concern regarding agents with multiple roles, however, I still feel that testing only on a two agent setup is a limiting assumption for a multi-agent exploration.

**Key Questions For Authors:**

Included in strengths and weaknesses section

**Limitations:**

Discussion of limitations is missing.

**Strengths And Weaknesses:**

## Strengths

- The authors provide a valuable extension to PARTNR by adding a dialog component, as well as a training split.
- I appreciated that the authors implemented their approach in an embodied-agent setting, rather than a text-based competition/collaboration game. This allows you to better understand whether the multi-agent methodology being proposed is complementary with larger environments and real-world considerations.
- The authors show that LlawCo can outperform or match approaches with 5x parameters, which indicates that conditioning on global behavior laws, distilled from prior behavior trajectories, is a parameter-efficient method for enabling centralized communication.

## Weaknesses

- The laws formulated in this paper seem to be global or team rules shared by all agents. This seems aligned with the centralized training, decentralized execution paradigm of collaboration. However, in many of these settings, each agent has a varying role or capabilities. To apply LLawCo in such settings, wouldn’t you need to train a different Law-aligned LLM for each agent? If so, this would be a significant limitation towards the large-scale applicability of such an approach.
- I wonder whether the fine-tuning stage for LlawCo is truly necessary. It is good to know that the fine-tuned version of the model performed 5% better than the base-model. However, I would argue that this gap can be closed through in-context techniques such as few-shot examples or RAG. Furthermore, there may be agentic or ensemble methods which can also be helpful to bridge this gap. Have the authors experimented with any such methods to gauge whether the fine-tuning stage is truly necessary?
- This paper does not provide any context regarding how the proposed approach will scale as the number of agents increase. Based on the paper, it seems as though there are always two agents in the benchmarks the authors test on. However, multi-agent collaboration is known to become more challenging as the number of agents increases, due to multi-threaded dialog and larger context management. How do the authors think their approach would scale as the number of agents grows?

---

> ### Author Rebuttal · Authors · 2026-03-31
>
> We thank the reviewer for recognizing the strengths of our work and for raising insightful concerns, which we address below.
>
> > Can LLawCo handle agents with different roles or capabilities using one shared law-aligned model, or would it require separate models for different agents and thus limit scalability?
>
> We thank the reviewer for this insightful question. LLawCo does not require a separate model for each agent. Instead, it learns a shared law-conditioned policy, where laws act as **high-level coordination priors** rather than agent-specific rules. Different behaviors emerge at inference time from each agent’s own context, including its observations, interaction history, and role or capability cues, without requiring separate models. This is also consistent with our experimental setting: based on PARTNR, PARTNR-Dialog already involves two heterogeneous agents (a human and a robot with different action spaces), and LLawCo works with a unified model. This is similar in spirit to parameter sharing in multi-agent RL, where one policy can produce different behaviors under different oservations. We will add this part of discussion in our paper.
>
> > Is the SFT stage truly necessary, or can stronger inference-time methods such as few-shot prompting, RAG, or agentic/ensemble approaches close the gap?
>
> We thank the reviewer for this thoughtful question. Our current ablation already tests the core comparison between in-context law usage and learned law alignment in **w/o SFT**. We believe the key difference is not only whether the model sees the laws, but whether it learns from **successful, law-aligned trajectories**. In LLawCo, the filtering and law-guided reflection stages construct training data that both solves the task and applies the laws correctly. In embodied multi-agent settings, such data is non-trivial, since failures often come from long-horizon coordination errors under partial observability.
> We agree that a stronger in-context baseline makes this comparison more complete. To test this, we add an enhanced inference-only baseline, **w/o SFT + RAG**, implemented following the ReAct-RAG design in the PARTNR paper. Specifically, during test time, the most relevant planning trace from the training dataset is selected based on sentence similarity and added to the LLM’s prompt. Our results below show that stronger in-context retrieval offer limited gains.
>
> | Setting | Laws in context | RAG | Param. update | Comp. | Succ. |
> |-|-|-|-|-:|-:|
> | Full LLawCo | Yes | No | Yes | 0.84 | 0.74 |
> | w/o SFT | Yes | No | No | 0.82 | 0.69 |
> | w/o SFT + RAG | Yes | Yes | No | 0.82 | 0.70 |
>
> Additionally, we view agentic or ensemble methods as orthogonal extensions rather than replacements for LLawCo, since they can be layered on top of our learned policy. We will add this result and part of the discussion in our paper.
>
> > How would LLawCo scale beyond the current two-agent setting, especially as larger teams introduce more complex dialog and context management?
>
> We thank the reviewer for this important question. Our experiments focus on the two-agent setting, which follows common practice in embodied collaboration benchmarks such as PARTNR, TDW-MAT, C-WAH, and dialog-based extensions like DialFRED. Most existing setups are built around two-agent interaction due to simulator and evaluation complexity. In particular, PARTNR itself is designed primarily to study controllable collaborative behavior, rather than to scale to a large number of agents.
>
> More importantly, LLawCo operates at the level of shared coordination principles rather than pairwise interactions. The learned laws are reusable across agents, and their complexity does not grow with the number of agents. In fact, as the number of agents increases and coordination becomes more challenging, such abstractions become more important.
>
> In practice, most components of LLawCo are independent of the number of agents, including law extraction, law-guided reflection, and SFT. The main part that grows is the dialog history. We list the percentage of each prompt component in the following table, communication typically accounts for only a limited portion of the total prompt (≈ 0.9%):
>
> | Component | Percentage (%) |
> |-|-:|
> | Instruction | 17.3 |
> | Possible Actions | 14.5 |
> | Observation History | 60.4 |
> | Action History | 5.1 |
> | Dialogue History | 0.9 |
> | Other | 1.8 |
>
> Additionally, scaling to more agents can provide richer interaction data with more diverse coordination failures. This can potentially in turn improve law induction, and the training pipeline can remain tractable through controllable filtering or sampling (e.g., filtering on higher completion rates or less steps). We view extending LLawCo to larger-scale multi-agent settings as a promising direction for future work.
>
> > Discussion of limitations
>
> We thank the reviewer for this suggestion. In the revision, We will add a limitations discussion covering safety, scalability, and training cost.

---

> > ### Author Rebuttal · Reviewer_pkxZ · 2026-04-08
> >
> > I thank the authors for their clarifications regarding varying agent roles and the relevance of fine-tuning. I have updated my score, but I still feel that conducting experiments only on a two-agent setup is limiting.

---

### Decision · Program_Chairs · 2026-04-30

**Decision:**

Accept (regular)

**Comment:**

The reviewers generally praise the self-improvement design requiring no stronger teacher model and the thoroughness of the ablation study. The main concerns centered on scalability beyond two agents, stability of induced laws across seeds, faithfulness of law-grounded reasoning, and potential self-confirmation bias. The author rebuttals have helped to address most of concerns, but still the scalability is unaddressed. So claims about broader multi-agent scalability remain somewhat speculative.

Overall, the paper appears technically solid, and likely to be useful to the embodied multi-agent community. Given the strengthened rebuttal and the final reviewer consensus, I lean accept, conditional on the authors incorporating all promised revisions. The authors are also encouraged to better frame scalability limits and safety caveats in the final version.